# Boosting functionality of synthetic DNA circuits with tailored deactivation

Kevin Montagne[1], Guillaume Gines[2], Teruo Fujii[2] & Yannick Rondelez[2,3]

Molecular programming takes advantage of synthetic nucleic acid biochemistry to assemble networks of reactions, *in vitro*, with the double goal of better understanding cellular regulation and providing information-processing capabilities to man-made chemical systems. The function of molecular circuits is deeply related to their topological structure, but dynamical features (rate laws) also play a critical role. Here we introduce a mechanism to tune the nonlinearities associated with individual nodes of a synthetic network. This mechanism is based on programming deactivation laws using dedicated saturable pathways. We demonstrate this approach through the conversion of a single-node homoeostatic network into a bistable and reversible switch. Furthermore, we prove its generality by adding new functions to the library of reported man-made molecular devices: a system with three addressable bits of memory, and the first DNA-encoded excitable circuit. Specific saturable deactivation pathways thus greatly enrich the functional capability of a given circuit topology.

[1] Department of Mechanical Engineering, University of Tokyo, Hongo 7-3-1, Bunkyo-ku, Tokyo 113-0033, Japan. [2] LIMMS/CNRS-IIS (UMI 2820), Institute of Industrial Science, The University of Tokyo, 4-6-1 Komaba, Meguro-ku, Tokyo 153-8505, Japan. [3] Gulliver UMR7083 CNRS, ESPCI Paris, PSL Research University, 10 rue Vauquelin, 75010 Paris, France. Correspondence and requests for materials should be addressed to Y.R. (email: yannick.rondelez@espci.fr).

DNA, as a readily available informational polymer with a well-documented chemistry and biochemistry, has proven instrumental to developing prototypal computers[1,2], controllers and circuits[3–7] based on molecules. These synthetic systems are inspired in part by cellular regulatory networks[8–10]. In turn, they can also be used as models to deepen our understanding or test hypotheses concerning biological circuits[11,12]. The basic conceptual components of those in vitro systems are: programmable molecular recognition based on Watson–Crick base-pairing; DNA-to-DNA catalysis (whereby some DNA species are able to regulate the kinetics of reactions involving other DNA species); and a straightforward interfacing with various physical[7,13–15], chemical[8,9,16,17] or biological[18–23] signals to be used as inputs, outputs or readouts[2,24,25].

Combining these elements, one can build networks made of chemical reactions, where each node is a particular DNA molecule or complex and where edges represent their kinetic relationships. As with electronic circuits, the function of these networks is deeply related to their topological properties. However, the precise kinetic laws of the chemical reactions connecting the network's nodes also play a fundamental role. In particular, the linear or nonlinear nature of these interactions are an essential determinant of the network's dynamics[10,26–28].

Of particular importance is a class of chemical programming languages based on dissipative processes that constantly consume energy[5,6,29,30]. Like electronic circuits or cellular signalling cascades, these molecular circuits can be reused, as they continuously recompute their answer to time-varying inputs. This key energetic feature opens the door to the programming of molecular self-organization[25,31] (as opposed to self-assembly), which is the driving force of countless biological feats[32,33]. The PEN DNA toolbox (Polymerase/Exonuclease/Nickase Dynamic Network Assembly toolbox) is a set of chemical reactions that has led to some advanced experimental demonstrations[30,31,34,35]. It is fuelled by dNTPs, based on enzymatic DNA polymerization/depolymerization steps and uses only two generic modules encoded by short single-stranded DNA templates (20–30 bases long): the first one, 'activation', mimics the basic stimulation of gene expression by a single transcription factor, while the second one, 'inhibition', emulates the converse, inhibitory process.

This toolbox combines experimental implementation protocols, generalized kinetic models and computational design tools. Short stable oligonucleotides (20–30 bases) are used as templates that instruct the topology of the network by physically connecting the sequence information of input compounds to that of output compounds. These templates encode the edges of a molecular reaction network where each node is a different short DNA molecule. Polymerizing/nicking cycles allow the input strand, acting as a trigger, to activate the generation of the signal strand encoded by the output side of the template. An exonuclease provides a degradation mechanism to maintain the system out of equilibrium. This versatile approach is based on mostly Michaelis–Menten production processes, where a single input is requested to produce outputs. As a consequence, rate laws are less-than-first order (that is, first-order saturable), and there is no direct and efficient mechanism to adjust the nonlinearity of the constituent interactions. Because of that, relatively large networks are required for certain behaviours. For example a four-node network was found necessary to produce a minimal robust bistable motif in PEN systems[4], whereas single-node, or two-node networks would be sufficient if one could adjust amplification thresholds[7,15,36]. And at least three nodes were found necessary for oscillations[5]. In contrast, biological systems rely heavily on nonlinear signal transduction—usually reflected in Hill coefficients larger than one in empirical rate laws—as a key ingredient of most networks and

dynamics. These nonlinearities result from cooperative behaviours (for example, multimeric regulation factors), delays (for example, because of transport between cellular compartments)[10], or other mechanisms such as zeroth-order ultrasensitivity[37] or sponging[38]. For example, the prototypical synthetic biological network, the repressilator, is critically dependent on the multimeric nature of the three transcription factors involved[28]. The benefit of nonlinearities for in vitro molecular programs was demonstrated with 'genelet' circuits, where an intermediate circuitry between the active elements can be harnessed to adjust thresholds. This titration-based approach led to compact bistable or oscillating networks[6,7,15].

Here we present a simple and efficient way to adjust the kinetic orders associated with individual elements of a template-based in vitro molecular circuit, by manipulating their degradation pathways. This is achieved by adding a saturable deactivating template to the PEN DNA toolbox. Using this extra tool, we show that second-order behaviours can be created simply and robustly, and demonstrate the new potentialities of the toolbox by designing a three-bit memory network and the first DNA-encoded excitable circuit.

## Results

**Tuning nonlinearites via degradation**. We started this analysis by considering a theoretical one-species network containing only a positive production feedback loop and a degradation pathway. In the absence of specific nonlinearities (for example, linear or Michaelis–Menten kinetics), this simple system provides at most a single stable steady state, whatever the respective rates (Fig. 1a–c). To obtain bistability, some curve twisting, that is, a change in the kinetic laws, is required: one may either tweak the shape of the production curve—to make it slower at low concentrations (Fig. 1d)—or adjust that of the degradation curve—to make it 'faster' at low concentrations (Fig. 1e). In both cases the goal is, graphically, to reveal three intersections between the production and degradation kinetic curves, two of these intersections being locally stable (that is, with higher production for slightly lower concentrations and higher degradation for slightly higher concentrations), and one being unstable.

We thus observe here that, in terms of function, we can reach the same goal by manipulating the linearity of either the 'production' process or the 'degradation' process. We note, however, that from a molecular engineering perspective, it may be easier to tweak the degradation pathway than to craft cooperative behaviours (with small enough leaks) from scratch. This is because a number of physical (for example, dilution) or chemical processes (for example, digestion and deactivation) with a variety of dynamical characteristics can be used as a sink. In the present case, the nonlinear 'degradation' rate law requested for bistability can be seen as the superposition of a simple first-order process and a saturable decay (Fig. 1f). From the graphical analysis, it is obvious that one requires the total first-order rate of degradation to be larger than that of production, while the maximum rate of the saturable degradation pathway should be smaller than the maximum production rate. In other words, the additional degradation path should be fast but have a small bottleneck (Supplementary Note 3.1).

**Building fast and saturable degradation pathways in vitro**. Figure 2 presents a molecular strategy to achieve the previous requirements. The approach is based on a 'drain' template, which elongates inputs very much like a regular DNA toolbox template, but the resulting duplex cannot be nicked. The elongated input slowly melts away, but its new 3′-tail, if correctly designed, now forbids any further priming on its cognate template—it has been

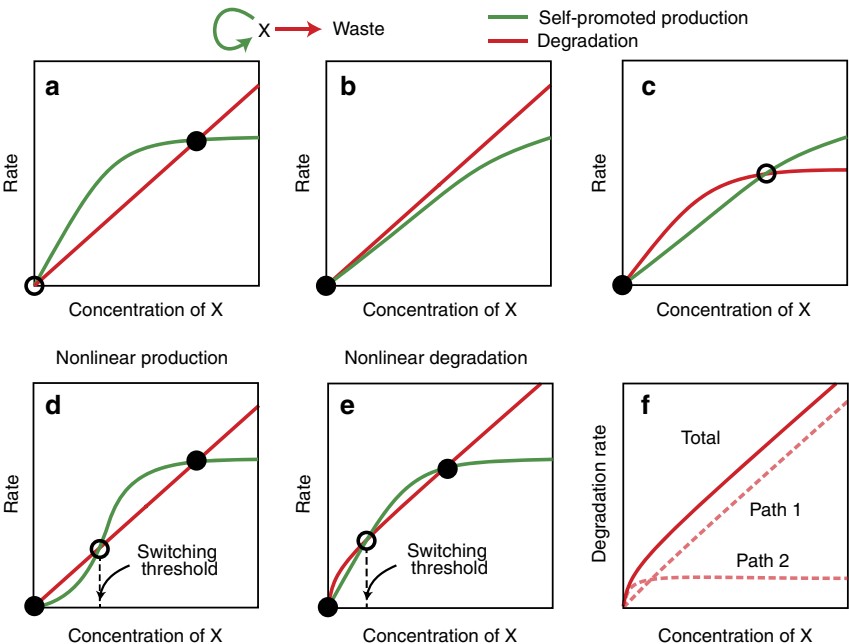

**Figure 1 | Stability and threshold in autocatalysis-degradation networks.** Species X is produced by an autocatalytic mechanism (green) and subject to degradation (red). (**a**–**c**) In the case of first-order or linear-saturable production and/or degradation, the system admits at most a single stable steady state (black circle), which can be null or not, depending on relative kinetic parameters (open circles indicate unstable fixed points). (**d**) In the case where production is of a higher order, slower production at low concentrations can stabilize the 0 state: the system is now bistable, with a threshold delineating the two basins of attraction. (**e**) The same holds with an 'increase' of the degradation rate at low concentrations. (**f**) The degradation curve required in case **e** can be constructed by combining two first-order/saturable degradation paths, one being faster and more saturated than the other.

wasted. The slow dehybridization step also restores the drain template, which therefore acts as a catalyst, not a substrate (like the amplification template, the drain template is protected against degradation by phosphorothioate modifications). Assuming enzymes are in their linear regime, the first-order rate associated with the drain pathway is controlled by the partial duplex formation—hence by drain concentration and binding affinity of the drain template relative to that of the amplification template. The latter can be adjusted, for example by removing a few bases on the binding site of the template. In turn, the maximum draining rate depends on the slow dehybridization step, and hence will be controlled by the stability increase brought by tailing the input along the drain template. We thus naturally define two thermodynamic design parameters, which are the binding energy difference between production templates and drain templates ($\Delta_{\text{bind}}$) and the increase in stability associated with extension on the drain template ($\Delta_{\text{ext}}$).

Because the set of enzymatic and DNA hybridization reactions involved in the DNA toolbox are well mapped, we can build a complete, quantitative kinetic model that keeps these two energy differences as free parameters (see Supplementary Note 2). This ordinary differential equation (ODE) model contains 15 variables, one for each DNA strand or possible DNA complex. A numerical search for bistability yields the following conclusions (Fig. 2e,f). First, bistability can be obtained over a wide range of the $\Delta_{\text{bind}}$/$\Delta_{\text{ext}}$ space, provided $\Delta_{\text{bind}}$ is above a minimal value, and enough drain templates are included. Second, drain templates with lower $\Delta_{\text{ext}}$ are more efficient. However, the requirement of fast capture of triggers imposes a minimal concentration of drain templates, which may sap the system if the drains' turnover is too fast (that is, low $\Delta_{\text{ext}}$). A sweet spot then appears around $\Delta_{\text{bind}} = 1$ kcal mol$^{-1}$ and $\Delta_{\text{ext}} = 3$ kcal mol$^{-1}$ (Supplementary Fig. 6), which we targeted for initial experiments (Table 1).

*In vitro*, we observed that in the absence of drain template, the simple, autocatalytic loop induced by a repeat template

(Fig. 2b) produces a first-order amplification with an intrinsically unstable 0 state: as already reported[39], even in the absence of trigger, the amplification eventually starts, initiated by spurious polymerization or impurities[40]. However, when a sufficient amount of drain template is incorporated in the system, we could observe that the 0 state becomes stable (Fig. 3a). At the same time, amplification could be readily observed after triggering with a few nanomolars of input strand (not shown here, see Fig. 6). We define Ct as the time the untriggered system takes to cross a given small fluorescence threshold (20% of the maximum signal amplitude). This delay is observed to diverge hyperbolically when increasing drain concentrations, in line with analytical or numerical predictions for a transition to bistability (Supplementary Note 3.1. and Supplementary Fig. 7). This behaviour is typical of a saddle node bifurcation, where bistability is suddenly observed above a critical concentration of drain. In this case, where $\Delta_{\text{bind}} = 1$ kcal mol$^{-1}$ and $\Delta_{\text{ext}} = 2.6$ kcal mol$^{-1}$, the critical drain concentration is found at ~17 nM.

We tested alternative designs with different $\Delta_{\text{bind}}$/$\Delta_{\text{ext}}$ values and found a rather good agreement with the predictions: drain templates require at least a two-base tail to efficiently deactivate the triggers; above this length, drain designs with lower $\Delta_{\text{ext}}$ are more efficient, that is, lead to bistability with lower concentrations (Fig. 3b; larger prototyping series are presented and discussed in Supplementary Note 1 and Supplementary Fig. 1). Still in line with predictions, a minimal value of $\Delta_{\text{bind}}$ appears necessary for bistable behaviour with reasonable drain concentrations, but excessive destabilization of the autocatalytic template yields unresponsive systems, even if they are very easily brought to the bistable regime (Fig. 3c). For a given template/drain pair in the bistable regime, the concentration of trigger required to switch the system to the high state increases with the concentration of drain template (Fig. 4). At the same time, the maximum amplification rate decreases with increasing drain concentrations as the latter starts outweighing the

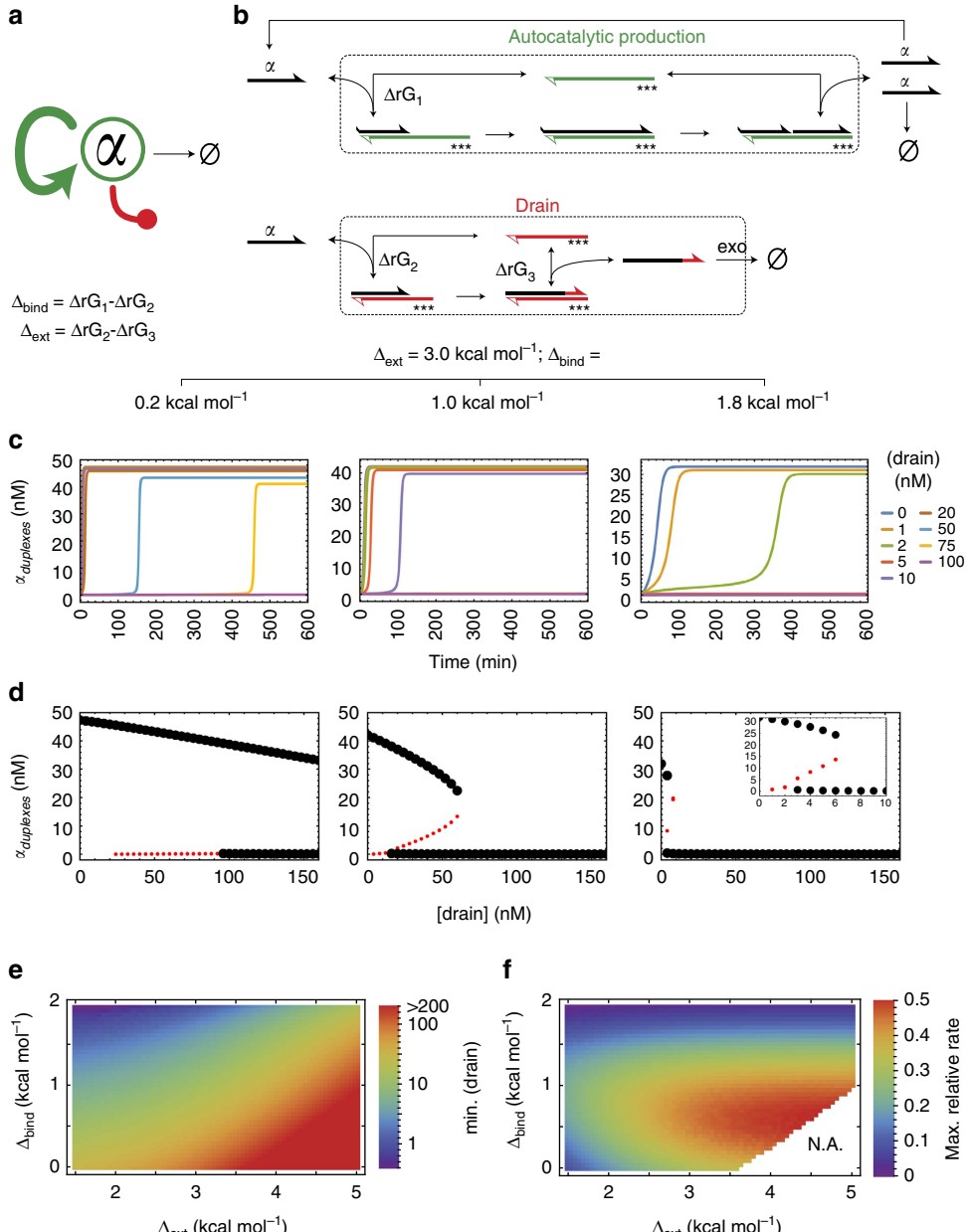

**Figure 2 | Drain-based bistability design for the PEN DNA toolbox. (a)** Schematic representation of a simple autocatalytic network with a sink and an additional decay pathway (dead-ended red arrow). The concentration of α is the dynamic variable considered here. **(b)** The species-specific and saturable degradation pathway is constructed from a short drain template that is able to bind the input α, extend it with a few additional bases and slowly release this deactivated species for eventual degradation. The difference between the binding strength on the drain and on the replication template ($\Delta_{bind}$) controls the respective reaction fluxes at low α concentration. The maximum throughput of the drain pathway depends on the dehybridization step, hence it is governed by the stabilization of the trigger upon extension on the drain ($\Delta_{ext}$). Empty arrowheads show non-extensible 3′-ends (3′ phosphate modification), and ··· denotes the protection against exonuclease degradation (backbone modification). **(c,d)** Predicted time traces and fixed points (stable in black, unstable in red), computed at $\Delta_{ext} = 3.0$ kcal mol$^{-1}$ using a complete kinetic model. The y axis corresponds to the total concentration of complexes involving α. Curves suggest that the bistable behaviour can be observed for a variety of $\Delta_{bind}$ values, if enough drain is present. Higher $\Delta_{bind}$ require less drain, but lead to sluggish amplifications and quickly fall back in a monostable trivial behaviour. **(e)** Computed concentrations of drain template necessary to reach the bistable regime in the $\Delta_{bind}/\Delta_{ext}$ parameter space (taken to reflect 0–3 deletions on the template and 2–6 bases on the drain tail). **(f)** Computed maximum amplification rates compatible with bistability (that is, at the minimal drain concentration). Values corresponding to drain concentrations above 200 nM were not computed.

positive feedback loop. In other words, increased robustness of the system is necessarily accompanied by a decrease in its responsiveness.

We also confirmed that it is possible to switch back from the high stable state to the low stable state by injecting a degradable inhibitor—a DNA strand that acts as a drain template, but is not protected against the exonuclease and thus has a short

lifetime in solution. In Fig. 5, we use fluorescence signals from both the unspecific intercalating dye, and a specific N-quenching modification attached to the template[25] to monitor unambiguously the state of the switch in real time. Repetitive injections of trigger γ—or degradable inhibitor drainγ′—show that this system can robustly maintain a single bit of memory as the high or low concentration level of a dynamic species, while

**Table 1 | DNA sequences used in this study.**

| Name | Sequence (5′–>3′) | $\Delta rG_1$ | $\Delta rG_2$ | $\Delta rG_3$ |
|---|---|---|---|---|
| *Trigger* | | | | |
| α | CATTCTGACGAG | | | |
| β | CATTCTGGACTG | | | |
| γ | CATTCAGGATCG | | | |
| ε | CATTCGGCCG | | | |
| | | | | |
| *Autocatalytic templates* | | | | |
| αtoα1 | C*T*C*GTCA**GAATGC**TCGTCAGAATG-P | − 10.2 | | |
| αtoα2 | C*T*C*G*TCA**GAATGC**TCGTCAGAAT-P | − 9.4 | | |
| αtoα3 | C*T*C*G*TCA**GAATGC**TCGTCAGAA-P | − 8.8 | | |
| αtoα4 | C*T*C*G*TCA**GAATGC**TCGTCAGA-P | − 8.0 | | |
| γtoγ | C*G*A*T*CCT**GAATGC**GATCCTGAA-P | − 8.8 | | |
| γtoγCy3.5 | Cy3.5*C*G*A*TCCT**GAATGC**GATCCTGAA-P | − 8.8 | | |
| βtoβ | C*A*G*T*CCA**GAATGC**AGTCCAGAA-P | − 8.7 | | |
| βtoβBMN3 | BMN3*C*A*G*TCCA**GAATGC**AGTCCAGAA-P | − 8.7 | | |
| εtoε | C*G*G*CC**GAATGC**GGCCGAATG-DY530 | − 10.7 | | |
| | | | | |
| *Drain templates* | | | | |
| drainα1 | A*A*A*A*CTCGTCAGAATG-P | | − 10.4 | − 13.0 |
| drainα2 | T*C*T*CGTCAGAATG-P | | − 10.1 | − 10.9 |
| drainα3 | T*T*C*TCGTCAGAATG-P | | − 10.1 | − 11.3 |
| drainα4 | T*T*T*CTCGTCAGAATG-P | | − 10.1 | − 12.0 |
| drainα5 | T*T*T*TCTCGTCAGAATG-P | | − 10.1 | − 12.8 |
| drainα6 | T*T*T*T*TCTCGTCAGAATG-P | | − 10.1 | − 13.5 |
| drainβ | T*T*T*T*TCAGTCCAGAATG-P | | − 9.9 | − 13.3 |
| drainγ | T*T*T*T*TCGATCCTGAATG-P | | − 10.1 | − 13.5 |
| drainε | A*A*A*ACGGCCGAATG-P | | − 11.0 | − 13.5 |
| drainγ′ | AACGATCCTGAATGA | | − 11.2 | − 12.4 |
| | | | | |
| *Decoys* | | | | |
| decoy1 | C*T*C*G*TCAGAATG-P | | − 10.4 | |
| decoy2 | C*T*C*G*TCAGAATGA-P | | − 10.4 | |

*indicates a phosphorothioate bond. The nicking enzyme recognition site on the template is in bold. P indicates 3′ phosphorylation. $\Delta rG_1$, $\Delta rG_2$ and $\Delta rG_3$, all in kcal mol$^{-1}$, are the binding free energies between, respectively, the autocatalytic template and a cognate trigger on its input side, the drain template (or decoy) and a trigger, and the drain template and an elongated trigger.

allowing multiple ON and OFF switching over 15 h (see Methods for details on fluorescence reporting and analysis).

**An apparent second-order amplification process**. Mathematically, it can be shown that, just above the critical concentration, the template/drain system should behave as a second-order autocatalytic system (Supplementary Note 3.1). Experimentally, the order of the reaction is difficult to assess directly from the shape of the time traces, because the fluorescence signal is a composite of many species' contributions. To extract the reaction order, it is more convenient to trigger the system with various initial concentrations $[x_i]$ of its input and record start time (Ct) values. Ct values will relate logarithmically with $[x_i]$ for a first-order amplification whereas an inverse law will reveal a second-order process (Supplementary Note 3.2 and Supplementary Fig. 8). Experimentally, in the absence of drain, we observed regular intervals between the amplification curves for a logarithmic range of initial trigger concentrations, therefore a first-order autocatalytic process. For a system just above the critical drain concentration, this pattern is disrupted and the Ct rather follows an inverse law, symptomatic of a second-order process (Fig. 6). Just below the critical drain concentration we see an intermediate case, which is indicative of an order between 1 and 2. Overall, this confirms that the drain approach, while based on the decay pathway, can be used to change the apparent kinetic law (not rate) of a self-activating positive feedback loop.

**Design of a three-bit synthetic memory network**. These results support the possibility of an efficient molecular programming by

tweaking degradation, rather than production, pathways. It is noteworthy that drains are intrinsically specific to their target. Because of the simplicity of the design, we thought that this modular approach could be applied to build larger systems. For example, bistability has been implemented *in vitro* from a number of molecular programming strategies[7,15], but so far, no synthetic design has been shown to have more than two stable states. In contrast, the presence of numerous attractors is believed to be an important feature of biological regulatory systems[8], and could explain for example the multipotency of stems cells[18,20,22]. In technological applications, networks with numerous stable states could be used to store and process multiple bits of digitalized information, for example in neural classifiers[2,24]. We thus attempted the construction of a system with four stable states, built from a mixture of two orthogonal one-bit molecular circuits of memory (Fig. 7a). We designed the sequences of the two autocatalytic subunits to avoid cross-binding, and the drain templates were constructed according to the rules introduced above. Relative concentrations were also inferred from the previous results, only keeping the total concentration of autocatalytic templates constant to mitigate enzyme load[26]. In addition, two fluorescent dyes were attached as N-quenching reporters[25] on the activation templates in order to generate species-specific fluorescence reporting.

When the four oligonucleotides were combined in a tube, the fluorescence signals indeed suggested four stable states ({0,0}, {0,high}, {high,0}, {high,high}), which neatly correlated with the presence or absence of the corresponding input in the initial mixture (Fig. 7b). The Cy3.5 signal (from γtoγCy3.5) was actually slightly lower once the β node was 'ON' than in the absence of

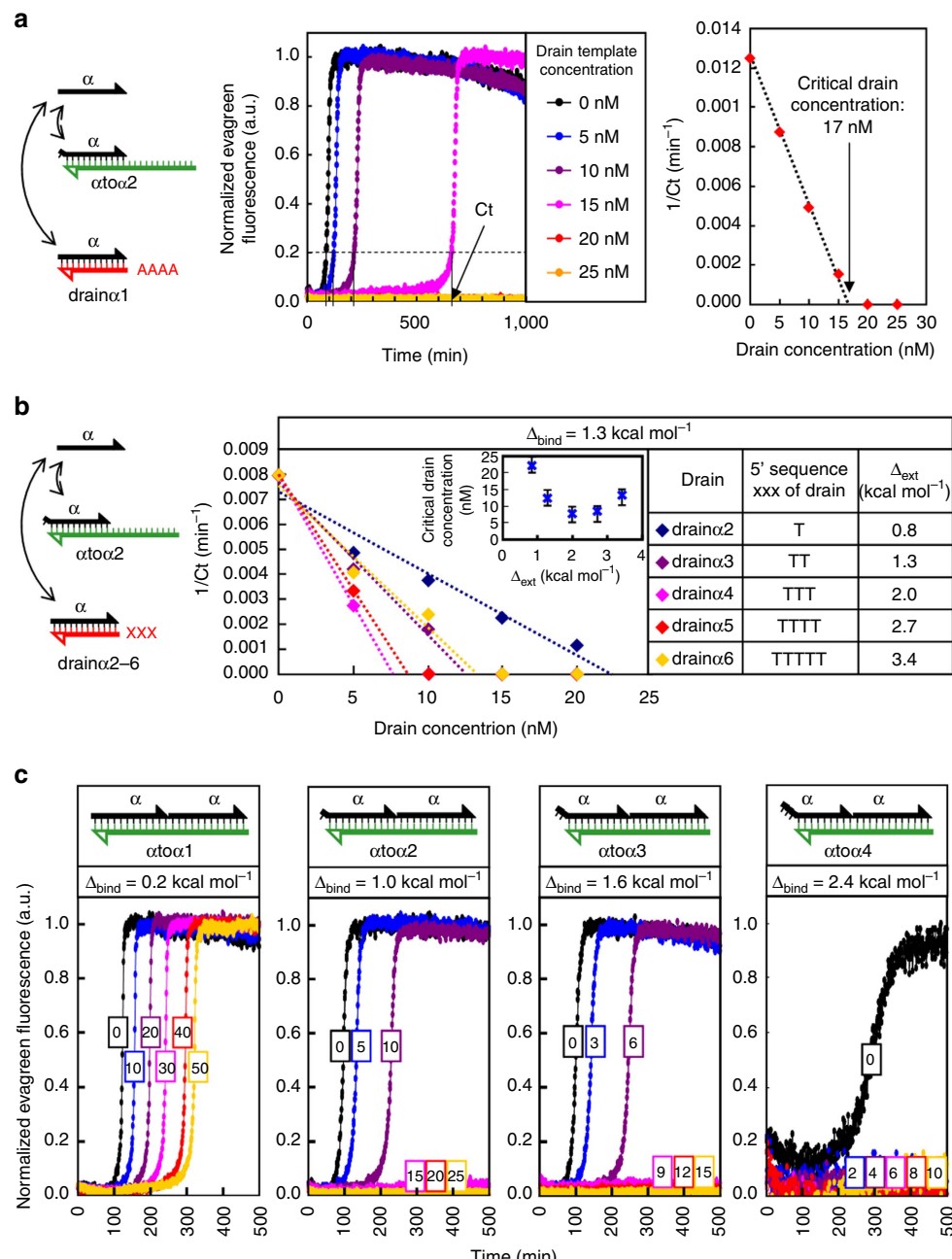

**Figure 3 | Experimental implementation.** Bst (polymerase), Nb.BsmI (nickase) and ttRecJ (exonuclease) are used to drive an autocatalytic template with or without drains. Fluorescence recordings using an intercalating dye reveal in real time the amplification process. (**a**) The template αtoα2 was incubated without trigger, in the presence of the indicated concentrations of drain template drainα1 (therefore $\Delta_{bind} = 1.0\,kcal\,mol^{-1}$ and $\Delta_{ext} = 2.6\,kcal\,mol^{-1}$). For each sample, the amplification delay (*Ct*, set as the time the fluorescence reaches 20% of its normalized maximum) is extracted. The simple autocatalytic network ([drain] = 0) always self-initiates, but increasing concentrations of drain template first delay, then completely abolish this phenomenon. The plot of 1/*Ct* versus [drain] reveals the hyperbolic transition to bistability and interpolation yields the critical drain concentration. (**b**) Effect of drain design. To check the influence of $\Delta_{ext}$ on the drain's inhibitory capacity, autocatalytic template αtoα3 was incubated with varying concentrations of various drains (drainα2 to drainα6). These drains all have the same sequence except for the three tailing bases at the 5′-end, therefore $\Delta_{ext}$ values range from 0.8 to 3.4 kcal mol⁻¹. The critical drain concentration is estimated by interpolation and reported in the inset with an error bar corresponding to the highest concentration of drain for which spontaneous amplification is observed and the first bistable point, respectively. (**c**) Influence of $\Delta_{bind}$ on the drain's inhibitory capacity. Autocatalytic templates αtoα1, αtoα2, αtoα3 or αtoα4 (schematized in the upper panels) were incubated in the presence of varying concentrations of drainα1 (concentrations indicated in the coloured boxes). $\Delta_{bind}$ ranges from 0.2 to 2.4 kcal mol⁻¹.

trigger, and reciprocally, the BMN3 signal (from βtoβBMN3) slightly shifted once the γ node was active. Ideally, the dye-labelled templates should respond solely to their cognate trigger. However, as both triggers used (β and γ) possess the same 5′-sequence for six nucleotides, it is possible that some reporting

crosstalk occurred in the presence of high concentrations of the other species.

Encouraged, we went on to a three-bit system by combining three drain-based bistable units in a single tube (Fig. 7c). The fluorescent signals in this case were not extremely clear for

                                                                    

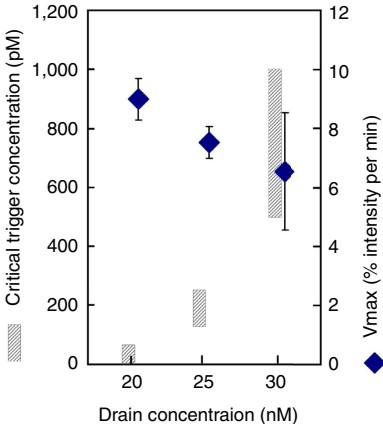

**Figure 4 | Tunable threshold can be implemented with drain templates at the cost of amplification rate:** 50 nM of autocatalytic template βtoβ was incubated with 20, 25 or 30 nM of drain template drainβ and various concentrations of trigger β; the hashed bars indicate the range, within which the threshold trigger concentration required to switch the system to the high state can be found; the blue dots and error bars, respectively, indicate the mean and s.d. of the maximum amplification rate Vmax measured in the presence of 1, 3 and 9 nM of trigger β.

the third element (possibly due to fluorescence overlaps), so we instead assayed the state of the system by aliquoting and using the isothermal real-time reamplification method previously reported[5]. Very much like quantitative PCR, this assay yields threshold-crossing $Ct$ values, which are logarithmically correlated to the concentration of the target species in the initial mixture and can be used to compute back its concentration. For each initial condition, we measured steady-state concentrations in the range 100–400 nM for the bits that had been switched ON, while untriggered bits yielded only trace values ($< 1$ nM). The differences in the levels of ON switches, from bit-to-bit and state-to-state, can be explained by the discrepancies in amplification/degradation efficiencies of the various sequences[41], as well as competition effects between switches. Overall, these results are consistent with the presence of eight distinct chemical states; hence, three independent memory bits accessible from the *ad-hoc* initial conditions (Fig. 7e).

**Combining with other regulatory motifs yields new functions.** Besides multistability, excitability is another fundamental feature of many biological processes involving pulses and transient behaviours[1,42]. It has been repetitively observed in non-equilibrium chemistry[43], but not yet rationally designed. An excitable circuit is a monostable system that responds to a stimulus with a long excursion, first amplifying the signal before coming back to rest to its ground state. It can be understood as the combination of a sensitive bistable module with a negative feedback loop. The latter will destabilize the high state and send it back to ground; therefore, small perturbations initiate large excursions in the phase space, before the system returns to its resting state. Following this reasoning, our approach to an excitable DNA system led us to insert the new drain-based bistable motif in a design providing a nonlinear negative feedback loop, using the predator-like autocatalytic mechanism previously described[3,13,14].

We use two fluorescent signals to measure individually the activities in the positive and negative loops. Applying perturbations of various intensities to this system, we indeed observe a critical drain concentration above which a strong

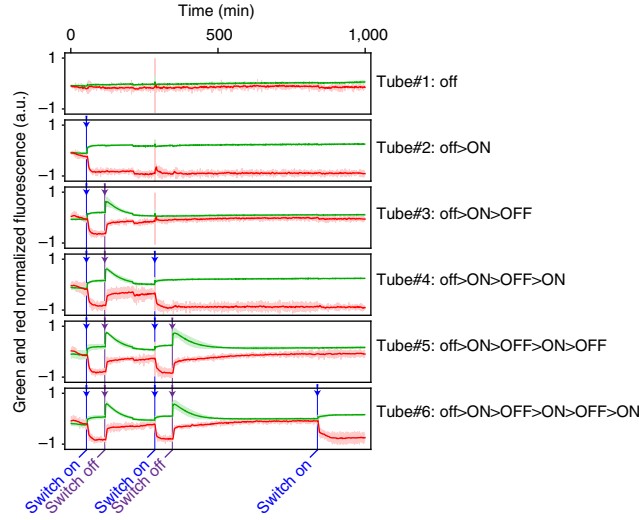

**Figure 5 | Reversible ON–OFF switching of a bistable system.** A series of replicate tubes containing the bistable system γtoγCy3.5/drainγ are prepared in the 0 state and repetitively switched ON—by addition of γ—and OFF—by addition of a degradable inhibitor (drainγ'). At each stage, all except one tube are actuated to prove the stability of the current state. The behaviour is monitored using both EvaGreen (total concentration of double-stranded DNA) and the red reporter Cy3.5 attached to the template, which shows a negative intensity shift when the template is in double-strand form[25]. Each trace is the mean ± 2 s.d. over three replicates.

response is generated first in the self-amplifying loop, then in the repression mechanism, before both fluorescence signals relax back to their initial values (Fig. 8). The drain template is necessary to trap the system in the 0 state and forbid the emergence of oscillatory cycles. Once the system has relaxed, a second supra-threshold perturbation induces a new large peak similar to the first one. At the same time, a sub-threshold injection of trigger fails to initiate the autocatalytic loop. Together with the two-dimensional phase plot of the excitable excursion, this shows that the system amplifies supra-threshold perturbations before coming back to its original (and unique) stable state, as expected for a genuine excitable network.

**Discussion**

We have shown that tinkering with the kinetic laws of degradation pathways can be a powerful and general approach to the molecular programming of functional circuits. While all experiments presented above used Nb.BsmI as the nickase of the PEN machinery, we obtained similar results with the other nicking enzyme used in reported PEN systems (Nt.BstNBI; see Supplementary Note 1.3 and Supplementary Figs 2–4).

It is particularly noteworthy that, while we only alter the linearity of the degradation pathway and in no way modify the one-to-one, first-order production mechanism, the outcome, from a phenomenological perspective, is the same as simply replacing a first-order production process ($A \rightarrow 2A$) by a second-order one ($2A \rightarrow 3A$). Indeed, we do observe an apparent second-order kinetic law for the amplification of the trigger. In addition, the excitable network shows that the approach is not only limited to the creation of bistable switches but also can be used to finely tune the nonlinearity associated with an amplifying node in a reaction network. Such nonlinearities are an essential determinant of the function of most network motifs[19,21,23,44]. In biological systems, they come in many cases from hard-coded

                     

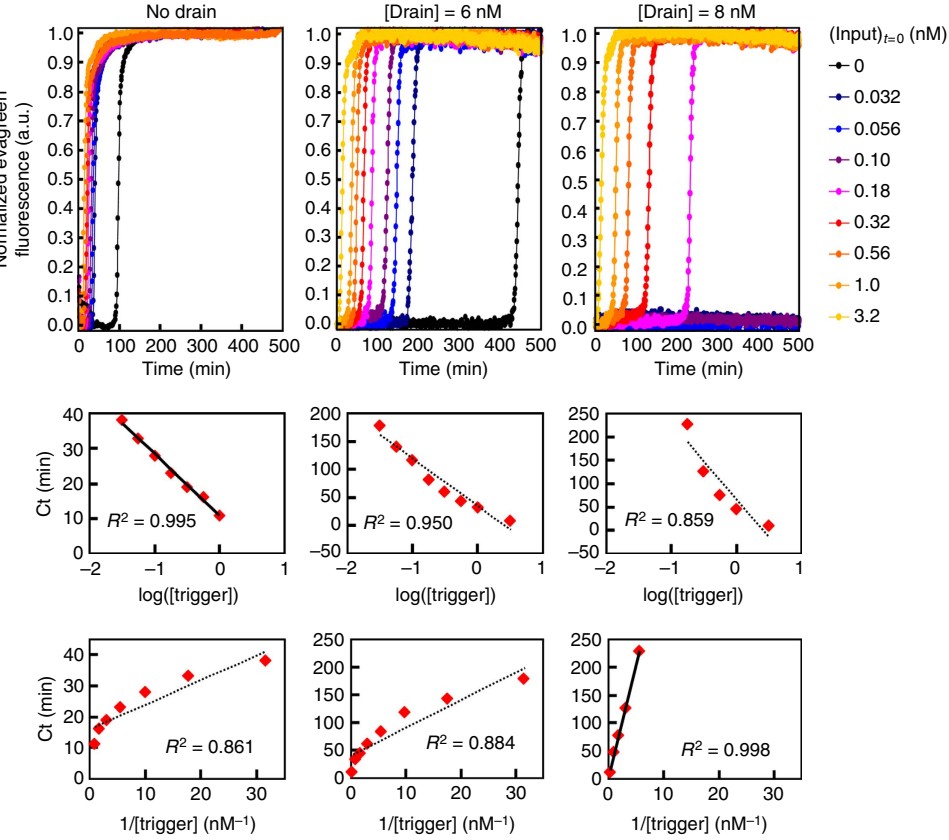

**Figure 6 | Assessment of the apparent amplification rate law.** A template/drain system ($\gamma$to$\gamma$/drain$\gamma$), where [template] = 50 nM and [drain] is at 0 nM, just below (6 nM) or just above (8 nM) the bistable point, is triggered with various initial amounts of input ($\gamma$). Time traces of the normalized fluorescence signal are shown on the top row. The existence of an amplification threshold is visible in the bistable area. Plots of the start time $Ct$ versus the initial concentration on a log scale (middle row) or an inverse scale (bottom row) are shown. A good linear fit is obtained on the log scale at drain concentrations below the critical one (0 and 6 nM), and on the inverse scale just above the bifurcation (8 nM drain). This implies respectively first- and second-order amplification for these systems close to the origin.

structural causes: a repressor dimerizes before it becomes active[25,45–47]; multisite phosphorylation events are required to activate a component[10,27,28,48]. However, functional approaches to nonlinear behaviours (like the one reported here) offer the advantage of being continuously tunable: the same components can be combined to produce either a monostable or a bistable switch. We could also imagine that the drain is dynamically produced by a control subnetwork, which would therefore drive the system between monostable and bistable regimes in response to other external signals.

Note that the mechanism presented here uses a form of competitive sequestration, but is qualitatively different from some other non-transcriptional regulatory mechanisms based on sequestration, because it involves an active modification of the dynamic species. For example, Burger *et al.* discuss the effect of a simple decoy transcription binding site, which tends to weaken bistability in a single gene network[6,29,30,49]. Competitive endogenous RNA are also a class of regulatory nucleic acids whose function is based on sponging, and which can give rise to ultrasensitive behaviour[31,50]. However, these are based on an *a priori* reversible binding of the target (even if it is possible that a variety of different mechanisms are involved, even for one given noncoding RNA[51]). The irreversible deactivation process described in the present work is thus closer to the targeted degradation involved, for example, in ubiquitination[30,52], or the nucleolytic action of some miRNA, which have indeed been shown to induce threshold effects[31,34,35,53,54]. In an open system, it has been shown theoretically[4,46] and experimentally[5,47] that the constant

production/dilution of a strong binder can also drive a positive feedback loop to bistability. However, this is not the case in a batch system as used here, because the strong binder would eventually get saturated. Not only this can be shown theoretically (Supplementary Note 3.3 on functional differences between various scenarios of branching pathways) but also can be experimentally demonstrated in the present molecular programming context: if the drain template is designed as a simple complementary sequence to the trigger, with no tail (that is, a 'passive' competitive binder, not involving a transformation of the substrate) we never detect bistability, even when the affinity of the trigger for the decoy was much larger than for the positive feedback loop template (that is, at large $\Delta_{bind}$; Fig. 9). Similarly, if the deactivation process becomes stoichiometric (no exonuclease), the transition to bistability is also lost (Supplementary Fig. 10).

We therefore see—experimentally here—that competitive side-branching in the network can produce very different functional outputs depending on the kinetic and contextual details. From a system-biological viewpoint, this should be taken as an additional motivation to finely evaluate both topological and kinetic effects in the study of biological networks[10,11]. In particular it suggests that bistability or nonlinear behaviour can be related to the characteristics of the decay processes, not only to the kinetics of the production processes.

On the quantitative side, the second important design element in such a 'post-transcriptional' programming approach is a precise control of the throughput (maximum rate) of the secondary degradation pathway (the drain). The effect of

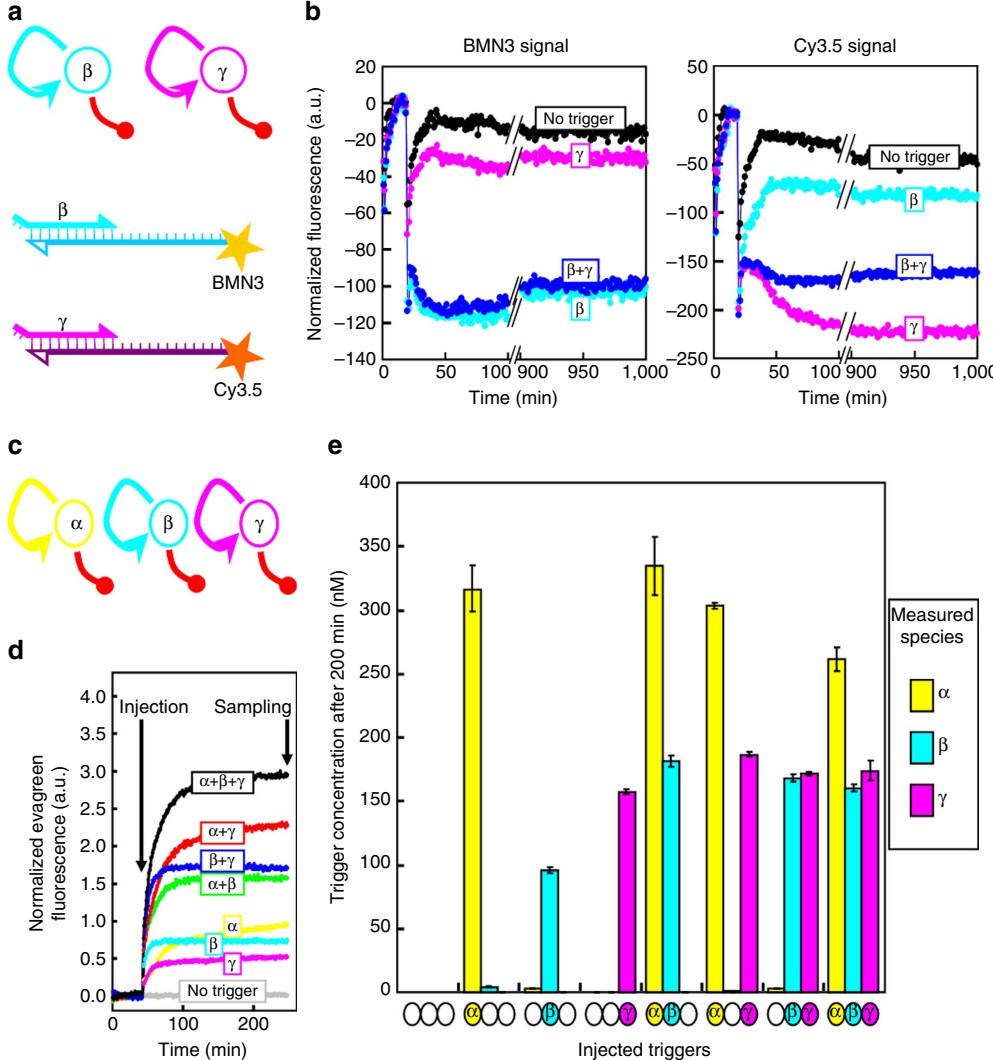

**Figure 7 | Multistable memory units.** (**a**) A two-bit system can be constructed by combining two bistable networks with orthogonal sequences, as schematized, and monitored with two fluorescent dyes attached to the templates[25]. (**b**) Change over time of BMN3 and Cy3.5 fluorescence after injection of the four possible combinations of the two triggers (no trigger; β only; γ only; β and γ). (**c**) Similarly, a three-bit system is obtained from three orthogonal nodes. (**d**) A mixture of αtoα3, βtoβ, γtoγ, drainα1, drainβ and drainγ was triggered with the eight possible combinations of their respective trigger (indicated in the coloured boxes) and EvaGreen fluorescence was monitored for 200 min. The nonspecific fluorescence signal suggests that steady states with one, two or three active nodes are attained. (**e**) The reaction in each tube from (**d**) was quenched after 200 min and the concentration of α, β and γ in the mixture were independently measured (see Methods). Only the nodes that had been triggered appeared to be in the high state, while untriggered ones were still in the low state. Each plot shows the mean concentration ± s.d. from two measurements.

nonlinear saturation behaviours in decay fluxes has recently attracted interest, mostly because it appears to be an elegant way to tune the functional robustness of a dynamical system[12,27,37,55,56]. Moreover, evidence is accumulating that this is indeed a biologically relevant regulation strategy[38,44]. For example, a degradation curve quite similar to the one described in Fig. 1e can be obtained from the combination of dilution (due to growth) and preferential degradation of monomers, for a dimerizing transcription factor[28,57].

In the present context of endowing bistability to an *in vitro* positive feedback circuit, the maximum rate at which the drain template can deactivate species is controlled by its concentration and the dehybridization rate of the duplex: both should be as low as possible to prevent the drain from outweighing the autocatalytic template. However, the first-order rate itself, also directly related to the concentration of the drain template, should be high. This leads to necessary compromises at the design stage. Once again, the versatility, fast prototyping

capabilities and model-friendliness of *in vitro* circuits are instrumental to explore these questions. We obtained a satisfying agreement between computational simulations and experimental results, suggesting that such approaches can also be embedded in a rational or computer-assisted[6,7,58,59] molecular programming strategy.

In future uses of species-specific drains, one may need to optimize different characteristics of the system, depending on the functional target. For example, when targeting larger systems in particular, one may want to limit the total amount of DNA templates in the reaction mix, hence work with drains that are effective at low concentrations. Alternatively, a bistable autocatalytic loop could be used as a sensitive detector of above-noise concentration fluctuations. In this case, one would rather minimize the size of the perturbation needed to cross the threshold and reach the high steady state, while maintaining the stability of the 0 steady state, leading to different design constraints. Since it may become difficult to link design options to the quality of the system with

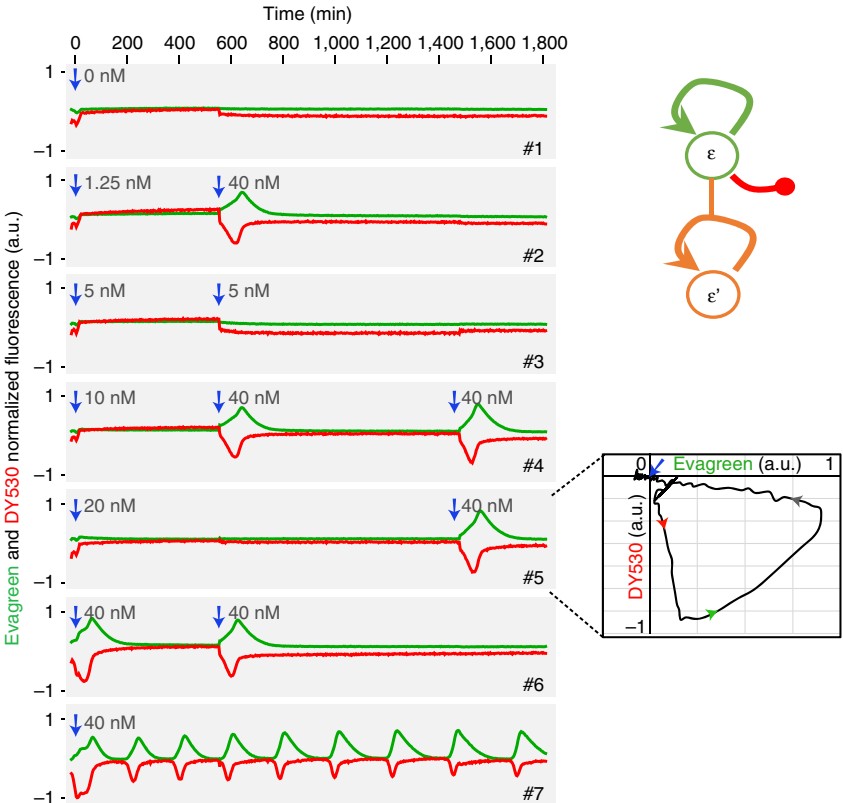

**Figure 8 | Excitable molecular network.** A positive feedback loop based on template εtoε is connected to the cognate drainε. The sequences are designed in such a way that the amplification process is also linked to a negative feedback reaction driving the consumption of triggers ε³. A series of replicates are initialized with different concentrations of trigger (from tubes #1 to #6: 0, 1.25, 5, 10, 20, 40 nM ε) and only the tube #6 shows a strong excitable response. At $t = 545$ and $t = 1,435$ min (blue arrows), triggers below (5 nM) or above (40 nM) the amplification threshold are injected in some of the tubes. Consistently, strong responses observed only above the threshold confirm the excitable character of the system. The inset displays the two-dimensional phase plot of the corresponding fluorescence traces, demonstrating the return to the 0 stable state after the excursion. The tube #7 contains only the template εtoε (no drain). It exhibits an oscillatory behaviour since the 0 state is unstable and cannot trap the system at the end of a cycle.

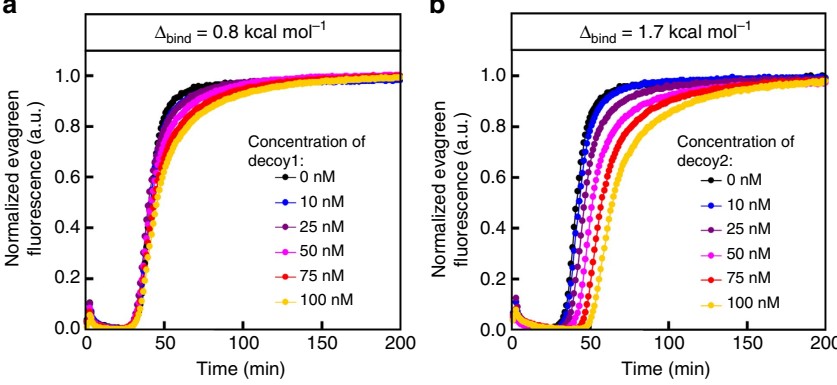

**Figure 9 | Effect of a simple decoy on a simple autocatalytic loop.** Autocatalytic amplification of template αtoα3 self-triggers within one hour and is marginally affected by the addition of decoy inhibitors decoy1 (**a**) or decoy2 (**b**), even when the decoy is a better binder and is twice more concentrated than the amount of autocatalytic template used (100 nM).

respect to its intended use, this warrants the introduction of automated design tools versatile enough to integrate additional, possibly untested regulatory options[58,59].

## Methods

**Oligonucleotides.** High-performance liquid chromatography-purified DNA oligonucleotides were obtained from either biomers.net or Integrated DNA Technologies. The sequences are listed in Table 1. Most autocatalytic templates and drain templates were modified at their 5′-end by phosphorothioate bonds and at their 3′-end by a phosphate group to respectively prevent their degradation by the exonuclease or their extension by the polymerase. An exception is the degradable inhibitor used in Fig. 5. This strand has the structure of a fast drain template, but was left unmodified, so that it could be cleared by the exonuclease, to allow multiple ON–OFF cycles. Templates βtoβBMN3, γtoγCy3.5 and εtoε were conjugated, respectively, to BMN3, Cy3.5 and DY-530 at their 5′-, 5′- and 3′-end (instead of the phosphate group), for fluorescent reporting. Thermodynamic parameters were calculated using the DINAMelt server.

**Reaction assembly.** Reactions were performed in a buffer containing 20 mM Tris-HCl, 10 mM (NH$_4$)$_2$SO$_4$, 10 mM KCl, 8.4 mM MgSO$_4$, 50 mM NaCl, 3 mM dithiothreitol (DTT), 2 μM Netropsin (Sigma-Aldrich), 100 μg ml$^{-1}$ BSA (New England Biolabs, NEB), 0.1% Synperonic F108 (Sigma-Aldrich), 1X EvaGreen dye (Biotium) and dNTPs (25 to 200 μM each). For most experiments, Bst DNA polymerase, large fragment and the nicking endonuclease Nb.BsmI (both from NEB) were respectively used at 8 and 400 U ml$^{-1}$ (respectively 0.1 and 4% final dilutions of the commercial stock solutions). For the excitable network, Bst DNA polymerase, full length (NEB) was used instead at a concentration of 40 U ml$^{-1}$ (0.8% of the stock solution). The thermophilic 5'->3' exonuclease from *Thermus thermophilus* ttRecJ was purified in the laboratory as previously described[60]. The stock solution was diluted to 1.34 μM in Diluent A (NEB) supplemented with 1% Triton X-100, and stored at −20 °C. It was used at a concentration of 25 nM throughout this study, except for the excitable network, for which 31 nM were used.

Reactions were assembled in a total volume of 20 μl and run at 45 °C in a MiniOpticon or CFX96 real-time PCR detection system (Biorad). For experiments requiring the injection of a DNA species, 0.5 μl of DNA solution was injected one or several times while the machine was paused for a minimal period; the final volume after all injections was also 20 μl. As reported for polymerase/nickase amplification schemes[39], we sometimes observed the emergence of template-independent amplification products after a long incubation time. These parasitic reactions—that may occur from the high or low state of the switches—are easily distinguished from their fluorescent signature and were not taken into account for the analysis.

Concentrations are:

Figure 3: (a) αtoα2 50 nM, drainα1 ranging from 0 to 25 nM. (b) αtoα3 50 nM; drainα2, drainα3, drainα4, drainα5 or drainα6, ranging from 0 to 20 nM. (c) αtoα1, αtoα2, αtoα3 or αtoα4 50 nM; drainα1 ranging from 0 to 50 nM.

Figure 4: βtoβ 50 nM, in the presence of drainβ 20, 25 or 30 nM, is triggered by β ranging from 0 to 9 nM.

Figure 5: γtoγCy3.5 25 nM and drainγ 10 nM are switched ON with 20 nM of γ and OFF with 60 nM of degradable drainγ'.

Figure 6: γtoγ 50 nM, in the presence of drainγ 0, 6 or 8 nM, is triggered by concentrations of γ ranging from 0 to 3.16 nM.

Figure 7: (a) βtoβBMN3 and γtoγCy3.5 25 nM each; drainβ and drainγ: 8 nM each; trigger combinations of 20 nM each of β and γ. (b) αtoα3, βtoβ and γtoγ: 17 nM each; drainα1, drainβ and drainγ: 15, 15 and 10 nM respectively; trigger combinations of 20 nM each of α, β and γ.

Figure 8: εtoε, 150 nM; drainε: 20 nM; triggered by various concentrations of ε ranging from 0 to 40 nM.

Figure 9 (a) αtoα3 50 nM; decoy1 ranging from 0 to 100 nM. (b) αtoα3 50 nM; decoy2 ranging from 0 to 100 nM.

**Data acquisition and analysis.** Reactions were monitored nonspecifically from the fluorescence signal produced by a double-stranded DNA intercalating dye (EvaGreen). In some cases, the templates were conjugated to additional fluorophores, such as BMN3 and Cy3.5. The signal of these fluorophores gets quenched when the 5'-end of the template is in a double-stranded form, providing a sequence-specific monitoring of the reaction in the system[25,40]. Fluorescence signals were acquired every minute (in some cases every 2 or 3 min) in the channels corresponding to the different dyes present in solution. ROX or Alexa-647-labelled dextran (Life Technologies) was used as a reference dye and added to the reaction for normalization, in order to mitigate the influence of signal artefacts during injection. Normalization was done by simply dividing the EvaGreen signal by the reference signal, after baseline removal; for clarity, the baseline was then set at 0 and the plateau (after amplification) at 1.

For quantification of the various species in the three-bit memory system (Fig. 7e), eight tubes containing the same three-bit mix, but triggered by various trigger combinations were allowed to run for 200 min before being frozen at −25 °C until needed. To quantify α, β and γ in the different tubes, three reactions were set up containing 50 nM of either αtoα3, βtoβ or γtoγ with, respectively, 9 nM of drainα1, 13 nM of drainβ or 6 nM of drainγ; 1 μl from each tube was then used to trigger each of the three reactions; the EvaGreen fluorescence was monitored and used to determine the Ct, which was then compared with Ct values obtained using a set of standard trigger solutions of known concentration.

**Data availability.** The data that support the findings of this study are available from the corresponding author upon request.

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

## Acknowledgements

We thank the members of the molecular programming group and the Fujii laboratories for helpful discussions and support. This work was supported by the JSPS Grant-in-Aid for Scientific Research on Innovative Areas 'Synthetic Biology' (number 23119001), and ERC Consolidator Grant 'ProFF' to Y.R. (number 647275).

## Author contributions

K.M. and G.G. designed, performed and analysed the experiments and wrote the manuscript. T.F. provided technical support and advice. Y.R. designed the study and performed preliminary experiments, analysed the results, conducted the mathematical modelling and wrote the manuscript.

## Additional information

**Competing financial interests:** All co-authors have submitted a patent application related to the work presented here.

**Publisher's note**: 

