## [Peer Review File · Nature Communications]

Reviewer #1 (Remarks to the Author)

Summary:

This paper describes an approach to obtain bistable behaviors tuning degradation rates in an artificial system of DNA based reactions using DNA polymerase for production, and Exonuclease and Nickase for degradation pathways (PEN). The system considered here is an autocatalytic DNA template, which amplifies a small trigger concentration via DNA polymerase. Degradation is mediated by Exonuclease, and modulated by changing the concentration of an active drain element, which sequesters trigger/output and makes it unsuited for autocatalysis (addition of a few bases) prior to its exonuclease degradation. By modulating the concentration of drain, the authors change the kinetic and stationary behavior of the system, achieving bistability in certain regimes.

Design tradeoffs for the thermodynamic interactions of input/trigger and template vs input/trigger and drain are thoroughly discussed. Mathematical modeling supports very nicely the experimental data. The authors demonstrate that their bistable system can be switched on or off multiple times, study how the autocatalytic reaction order changes in the presence of the drain, and finally test the behavior of multiple switches operating in parallel.

Main strengths:

- The approach is elegant and experiments are very well matched by modeling. These results are certainly appealing to interdisciplinary audiences.
- The discussion of thermodynamic tradeoffs between input to template and input to drain binding pathways is thorough and useful.

Main general weaknesses:

- Although the whole project is nicely put together, I would not say that tuning degradation rates to determine dynamic/static behaviors in a network is a majorly new idea. This might be one of the few papers to spell the idea out, but this principle is routinely used in synthetic biology.
- The PEN toolbox is not new, and although a significant amount of labor went into these experiments, the overall contribution seems incremental. A major issue with the PEN toolbox is that it is largely incompatible with other DNA nanotechnology systems, and it is unclear what is its usefulness, beyond the synthesis of molecular reaction networks for proof of principle.

Specific comments:

- The paper starts off suggesting the following recipe: to obtain bistable behavior it is sufficient to design a system with 3 steady states. This recipe is misleading, because this is not true in general. The presence of 3 steady states guarantees bistability only if the system satisfies other assumptions, such as monotonicity and dissipativity - see, for example, Angeli, Ferrell and Sontag, PNAS 2004. It is unlikely that this system satisfies such assumptions, so the presence of 3 steady states does not automatically imply bistability. (Bistability might just be a lucky coincidence.)
- Demonstration that this bistable system can be toggled between states is the most interesting contribution of this paper, as far as I am concerned. Unfortunately, this section is somewhat poorly written. There is mention of a degradable inhibitor, but there is no explanation of what additional reaction this is. It would be very helpful to mention that reporter systems and normalization procedures are described in the Methods section at the end of the paper.
- Multistable memory units: the way these results are purported is controversial. Essentially, this is a demonstration that one can operate together several individual bistable circuits that are functionally disconnected from each other. I would not call this a multistable system, rather an array of bistable systems. These bistable circuits may be coupled indirectly because they compete for enzymatic resources or by unwanted binding of strands (cross-talk), but it looks like these are not significant problems. Accurate sequence design to avoid crosstalk is routinely done in DNA nanotechnology, with demonstrations of dozens of components operating together.

A better way to spin the importance of these results is that they indicate that it is possible to build a 2 bit or a 3 bit system, which - based on the results at Fig. 5, might be reversibly switched.

- In the discussion section, the authors observe that a decoy strand that simply sequesters α (or input, referring to Fig. 2) without degrading it is not sufficient to obtain bistability, although it (unsurprisingly) introduces delay. Well, this decoy can't really be compared with a degradation rate, so I am not sure how interesting this is.

- What is the contribution of the drain/sink in terms of competition for DNA polymerase? What if in practice bistability was also due to the fact that production decreases due to competition between drain and template?

A reduction in free DNA polymerase concentration could be consistent with the transition to a second order rate-like behavior; because there is less DNA polymerase available (some of it is wasted to produce inert copies of α from the drain), then two copies of α are required for a net increase of one copy of α . If my reasoning is correct, then it might be misleading to claim that bistability was obtained by exclusively tuning the degradation rates in the system. Because of the way the system is built, there may be an indirect change in the production rates as well.

- At page 8, I found the approach to derive information on rate order from delay to be interesting. However, I was confused by the reasoning "the order of the reaction is difficult to assess directly from the shape of the time traces, because the fluorescence signal is a composite of many species contributions". Wouldn't the delay also be a quantity that reflects the behavior of whole sample, because it is determined from that same curve that "is a composite of many species contributions"? I can buy the idea that initially the drain reaction dominates before autocatalysis kicks in, so by assessing the time delay it takes before autocatalysis goes off is an indirect readout of the kinetics of the drain/input dynamics. However, I suggest that these paragraphs are rewritten.

As for the conclusion of this paragraph, it seems to me rather obvious that by introducing new reactions one does not only change the stationary behavior of a system, but also its kinetics.

- The paper needs to be read more carefully, to make sure that the necessary information is provided timely and accurately to the reader. I found several omitted definitions, and unclear sentences; for example, at the top of p 8 there is a sentence "results obtained with the other nicking enzyme used in PEN systems (Nt.BstNBI)", that seems completely disconnected from the rest; so far no specific enzyme was mentioned in the main text - prompting the reader to confusion.

Fig 2 and the main text should explain what enzymes are being used; what are the system inputs and what are its outputs or readouts. The paragraph in the middle of page 2 (introduction) describing the PEN toolbox is too vague and requires readers to look up earlier papers.

The time C_t it takes for the system to spontaneously switch steady state from low to high is used early in the paper (Fig 2), but it is only loosely defined at the end of page 8. Please provide a clear explanation of how this time was computed.

- The supplement should be cleaned up and better connected to the main paper. References in the supplement are not compiled and all appear as question marks.

Reviewer #2 (Remarks to the Author)

The authors described a simple and versatile method to change the nonlinearity of the network by regulating the degradation kinetics and created a bistable switch without changing the network topology.

In addition, they showed several possible systems by using this bistable switch.

The proposed systems (i.e. memory, excitable molecular network) lacks novelty and usefulness because the systems in and of themselves have been already reported and authors did not show real world applications.

However, the concept that the linearity can be changed by tuning the degradation kinetics without changing of the network topology is a significant idea for extending the network functions. Thus, this manuscript seems to be sufficient to publish in Nature Communications if the idea has not been reported yet.

Authors should address the following questions and concerns.

Question 1:

Authors describe "see SI Appendix for a mathematical discussion" in page 5, but do not show section number or figure number.

Authors should point where the discussion about "additional degradation path" and "small bottleneck" is in "SI appendix for mathematical discussion".

Question 2:

Authors draw regression lines in figure 3-B. However, the number of measurement point is too few,

for example, the regression line of drain-alpha 4 seems to be calculated by using single measurement value.

Authors should measure the $1/C_t$ at more than three drain concentrations for each line in order to verify that the values of $1/C_t$ are lined on the straight lines.

In addition, authors should add error bars in all plots.

Question 3:

In figure 5, authors use dye-labeled template to measure the ON and OFF state.

However, ON and OFF state should be defined by the amount of produced DNAs rather than template duplex.

So, authors should show the relationship between the fluorescent signals and the amount of the product

by direct quantification of the product by other method like qPCR.

Question 4:

Author describe that two independent bistable switches can make four stable states in figure 7-B. I expect that "no trigger" and "beta" shows equivalent Cy3.5 signal because the "beta" does not interact with the gamma-template which is labeled with Cy3.5.

Similarly, the value of the beta bistable switch in figure 7-E increases in the presence of other triggers. I'm confused because authors describe "independent" but the results seems not to be independent.

Authors should mention about this conflict.

Question 5:

Although the delta Gs of the drain templates in table 1 seems to be almost the same,

why the trigger concentrations after 200 min are differ in alpha, beta and gamma in figure 7-E?

According to authors' claim, I expect that the same binding affinity results in the same kinetics of products, or the same amount of products. Author should explain why the amount of the products differs

and experimentally verified the explanation.

Question 6:

Authors should add an illustration of the network to figure 8 like figure 3-A or figure 2-B for reader's help to understand the network in detail.

In addition, authors should add and compare a control which does not include non-linear system to show the importance of the bistable (non-linear) switch for this excitable molecular network.

Reviewer #1 (Remarks to the Author):

Summary:

This paper describes an approach to obtain bistable behaviors tuning degradation rates in an artificial system of DNA based reactions using DNA polymerase for production, and Exonuclease and Nickase for degradation pathways (PEN). The system considered here is an autocatalytic DNA template, which amplifies a small trigger concentration via DNA polymerase. Degradation is mediated by Exonuclease, and modulated by changing the concentration of an active drain element, which sequesters trigger/output and makes it unsuited for autocatalysis (addition of a few bases) prior to its exonuclease degradation. By modulating the concentration of drain, the authors change the kinetic and stationary behavior of the system, achieving bistability in certain regimes.

Design tradeoffs for the thermodynamic interactions of input/trigger and template vs input/trigger and drain are thoroughly discussed. Mathematical modeling supports very nicely the experimental data. The authors demonstrate that their bistable system can be switched on or off multiple times, study how the autocatalytic reaction order changes in the presence of the drain, and finally test the behavior of multiple switches operating in parallel.

Main strengths:

- The approach is elegant and experiments are very well matched by modeling. These results are certainly appealing to interdisciplinary audiences.*
- The discussion of thermodynamic tradeoffs between input to template and input to drain binding pathways is thorough and useful.*

Main general weaknesses:

- Although the whole project is nicely put together, I would not say that tuning degradation rates to determine dynamic/static behaviors in a network is a majorly new idea. This might be one of the few papers to spell the idea out, but this principle is routinely used in synthetic biology.*
 - The PEN toolbox is not new, and although a significant amount of labor went into these experiments, the overall contribution seems incremental. A major issue with the PEN toolbox is that it is largely incompatible with other DNA nanotechnology systems, and it is unclear what is its usefulness, beyond the synthesis of molecular reaction networks for proof of principle.*
-

We wish to thank Reviewer#1 for the positive appreciations (and for the many useful suggestions listed below), but we kindly disagree with the last two statements.

* First, as explained below, we do not merely tune the degradation kinetic **rate**. This would never drag the system into a bistable regime, because tuning the rate does not alter the linearity/nonlinearity of the behavior (note that of course, it is trivial to tune degradation **rates** in our in vitro experiments, by simply adjusting the exonuclease concentration. We do that routinely for the fine-tuning of new circuits).

Instead, in this report, we explain that tuning the **shape** of the degradation curve, (i.e. the kinetic **law**, not the kinetic **rate**) is much more powerful, and we precisely demonstrate a way to do that in the context of the PEN toolbox. In this regard, as rightly noted below by the reviewer, Fig. 6 is a very important part of the paper, as it demonstrates that crafting the degradation kinetic **law** allows to emulate second order **production** dynamics (something that could obviously not be achieved by playing only with the degradation **rate**).

Indeed, synthetic biology sometimes makes use of degradation tags (such as *ssrA*) to accelerate turnover of protein components by directing them to efficient endogenous or exogenous proteases (and in some cases this may have unexpected nonlinear consequences as detailed in my PRL 2012 paper ¹), but this is typically seen merely as a way to balance production and degradation rates. On the contrary, our approach to compensate for featureless **production** dynamics by constructing ad hoc “bumpy” and species-specific degradation curves is a rational, new and general strategy to achieve non-trivial dynamics, e.g. multistability or excitability as demonstrated in the paper.

* Regarding the compatibility and usefulness of the PEN toolbox, we think that the conclusion of the reviewer is a bit hurried, as explained below. In any case, we think that this personal opinion should not directly cause the rejection of this manuscript, which, in addition to introducing a new efficient tool for PEN systems, has the more universal merit of bringing the attention to the degradation pathway as a powerful approach to craft interesting circuits and manage nonlinearities.

One has to note that the PEN toolbox is still a relative newcomer to the field of Molecular Programming (with a first paper in 2011), whereas competing approaches have a much longer history (e.g. toehold mediated strand displacement originated in 2000, and genelets in 2004). True, we have spent the last five years solving many “proof of principle” challenges in the design of synthetic dynamics. But, together with other groups, we are now very much involved in taking advantage of the robustness/versatility of this system to design practical applications. [redacted]

The remark on compatibility is also undeserved in our opinion. Compatibility is tricky for any molecular well-mixed system, but DNA-based approaches have a clear strength in this respect, because DNA can connect to a huge range of other chemistries or physical processes. And

indeed, the PEN toolbox is doing quite well in this respect. [redacted] For example Team Sendai at BIOMOD2014 had a functional system (now an oral presentation at DNA22, see <http://www.dna-node.com/dna22/accepted-papers-and-abstracts/index.html>) that uses a combination of PEN DNA and toehold-mediated strand displacement.

We hope these incomplete arguments will convince the reviewer#1 to reserve its conclusion on PEN for a few more years, as we believe that many exciting applications are coming (in fact, most of the ones we are aware about actually use the new “drain” strategy reported in the present manuscript)

Specific comments:

- The paper starts off suggesting the following recipe: to obtain bistable behavior it is sufficient to design a system with 3 steady states. This recipe is misleading, because this is not true in general. The presence of 3 steady states guarantees bistability only if the system satisfies other assumptions, such as monotonicity and dissipativity - see, for example, Angeli, Ferrell and Sontag, PNAS 2004. It is unlikely that this system satisfies such assumptions, so the presence of 3 steady states does not automatically imply bistability. (Bistability might just be a lucky coincidence.)

Of course we did not aim to imply sufficiency in the general case, and we thank the reviewer for pointing that the manuscript was unclear in this respect. Our discussion is strictly restricted to the 1D case (single variable ODE) where it is easy to demonstrate that steady states need to alternate their stability (briefly, the demonstration goes as: let f correspond to the righthand side of the differential equation describing the dynamics, and points $f(x) = 0$ be steady states. If f is smooth on interval $[a,b]$, f is equal to 0 only in a and b and $f'(a)$ and $f'(b)$ are nonzero (non-degeneration of the steady state), then $f'(a)$ and $f'(b)$ must have different signs. Therefore non-degenerate steady states must alternate their stability as we go along the line or the interval. If two states are stable, there is one unstable state in-between).

We have changed the text (including the abstract) to clarify this and also used the opportunity to emphasize the difference between tuning rates (Fig 1A, B, C) vs changing laws (fig D, E):

- We added: “Species X is produced by an autocatalytic mechanism (green) and subject to degradation (red).” in Fig caption 1
- We also changed: “We started this analysis by considering a theoretical network containing only a positive production feedback loop and a degradation pathway. In the

absence of specific nonlinearities, this simple system provides at most a single stable steady state (Fig. 1). To obtain bistability, some curve twisting (i.e. change in the kinetic law) is required: one may either tweak the shape of the production curve -to make it slower at low concentrations (Fig. 1D)- or adjust that of the degradation curve -to make it *faster* at low concentrations (Fig. 1E). “

To: “We started this analysis by considering a theoretical **one-species** network containing only a positive production feedback loop and a degradation pathway. In the absence of specific nonlinearities (**i.e. linear or Michaelis-Menten kinetics**), this simple system provides at most a single stable steady state, **whatever the respective rates, Fig. 1A,B,C**. To obtain bistability, some curve twisting, **that is, a change in the kinetic laws**, is required: one may either tweak the shape of the production curve -to make it *slower* at low concentrations (**Fig. 1D**)- or adjust that of the degradation curve -to make it *faster* at low concentrations (**Fig. 1E**). “

- Demonstration that this bistable system can be toggled between states is the most interesting contribution of this paper, as far as I am concerned. Unfortunately, this sections is somewhat poorly written. There is mention of a degradable inhibitor, but there is no explanation of what additional reaction this is. It would be very helpful to mention that reporter systems and normalization procedures are described in the Methods section at the end of the paper.

Thank you for the positive comment. We have improved this part using these recommendations. The Methods section is now mentioned (and has been reorganized). The new paragraph is as follows:

We also confirmed that it was possible to switch back from the high stable state to the low stable state by injecting a degradable inhibitor –**a DNA strand that acts as a drain, but is not protected against the exonuclease and thus has a short lifetime in the solution**. In Fig. 5, we use **fluorescence signals from both the unspecific intercalating dye, and a specific N-quenching modification attached to the template to monitor unambiguously the state of the switch in real time**. Repetitive injection of trigger of γ -or degradable inhibitor drainy’- **shows that this system can robustly maintain a single bit of memory as the high or low concentration level of a dynamic species, while allowing multiple ON and OFF switching over 15 hours (see Methods for details on fluorescence reporting and analysis)**.

The figure caption gives additional details: “A series of replicate tubes containing the bistable system γ to γ Cy3.5/drainy are prepared in the 0 state and repetitively switched ON -by addition of γ , and OFF -by addition of a degradable inhibitor (drainy’). At each stage, all except one tube are actuated to prove the stability of the current state. The behavior is monitored using both EvaGreen (total concentration of double-stranded DNA) and the red reporter Cy3.5 attached to the template, **which shows a negative intensity shift when the template is in double strand form²⁵.**”

For coherency, we now also mention the protection of drain templates in the paragraph **Experimentally building fast and saturable degradation pathways:** “**like the amplification template, the drain template is protected against degradation by phosphorothioate modifications**”.

- Multistable memory units: the way these results are purported is controversial. Essentially, this is a demonstration that one can operate together several individual bistable circuits that are functionally disconnected from each other. I would not call this a multistable system, rather an array of bistable systems. These bistable circuits may be coupled indirectly because they compete for enzymatic resources or by unwanted binding of strands (cross-talk), but it looks like these are not significant problems. Accurate sequence design to avoid crosstalk is routinely done in DNA nanotechnology, with demonstrations of dozens of components operating together. A better way to spin the importance of these results is that they indicate that it is possible to build a 2 bit or a 3 bit system, which - based on the results at Fig. 5, might be reversibly switched.

We have followed this suggestion. We changed our terminology to “n-bit system”, and mention the overall number of stable states. Concerning the competition for enzymatic resources, it is indeed a very general feature of enzyme-driven networks, but as the reviewer notes, it does not forbid the building of such multinode systems, given that appropriate measures are taken (as we mention in the text, we simply kept: “the total concentration of autocatalytic templates constant to mitigate enzyme load²”, and this strategy worked).

- In the discussion section, the authors observe that a decoy strand that simply sequesters λ alpha (or input, referring to Fig. 2) without degrading it is not sufficient to obtain bistability, although it (unsurprisingly) introduces delay. Well, this decoy can't really be compared with a degradation rate, so I am not sure how interesting this is.

Here we wanted to show that it is indeed the de-activation of alpha that is important, and not just its reversible sequestration. We have noted that the literature is not always very precise about the difference between reversible, irreversible (suicide) and irreversible catalytic (the present case) competing pathways, and this section is an attempt to clarify these points using the present synthetic circuits as examples. We show here that the reversible competitor can never bring the system in a bistable regime, even if it is a much better binder than the autocatalytic template. To complete this part and make it more coherent, we have added in SI an experiment showing the irreversible suicide case. This is done by simply omitting the exonuclease, which is the final sink in this system. The new Fig. S11 clearly shows that in this case, the sharp hyperbolic bifurcation to the bistable regime is lost, as irreversible competitors simply delay more and more the amplification. All these results are in agreement with the mathematical models presented in SI Note 3.3 (SI is now better called throughout the text)

Changes:

Added on page 15: “(Supplementary Note 3.3 on functional differences between various scenarios of branching pathways)”

And: “Similarly, if the deactivation process becomes stoichiometric (no exonuclease), the transition to bistability is also lost (Supplementary Fig. S11).”

- What is the contribution of the drain/sink in terms of competition for DNA polymerase? What if in practice bistability was also due to the fact that production decreases due to competition between drain and template?

A reduction in free DNA polymerase concentration could be consistent with the transition to a second order rate-like behavior; because there is less DNA polymerase available (some of it is wasted to produce inert copies of α from the drain), then two copies of α are required for a net increase of one copy of α . If my reasoning is correct, then it might be misleading to claim that bistability was obtained by exclusively tuning the degradation rates in the system. Because of the way the system is built, there may be an indirect change in the production rates as well.

As can be seen in Fig. 2, the basin of attraction of the null state is typically very small, which implies that departure from this state is a phenomenon that happens at low trigger concentration. Hence it is controlled by the linear regime of the enzymes, and saturation/competition is not critical here.

Note also that the transition to bistability is obtained not by changing the rate, but by changing the shape of production or degradation curves. In the case of production one would need to lower the production rate ONLY at low concentrations, and it is not clear how this could be obtained by polymerase competition only (if the competitive substrate for the polymerase has no relation to the autocatalytic species).

The new experiments in Fig. S11 confirm these arguments, as it shows that inclusion of a competitive suicide inhibitor (which also loads the polymerase) does not lead to bistability.

As a side note, once again, we never claimed that “*bistability was obtained by exclusively tuning the degradation rates in the system*”. On the contrary, we emphasize the need to adjust the kinetic laws of the degradation pathway.

To clarify this point, we added “**Finally note that since the basin of attraction of the null state is typically small (see Fig. 2 in the main text), departure from this state is a phenomenon that involves low trigger concentration. Hence it is controlled by the linear regime of the enzymes, and we do not expect polymerase competition (between the template and the drain template) to play a significant role in the transition to bistability.**” in SI note 1.1.

- At page 8, I found the approach to derive information on rate order from delay to be interesting. However, I was confused by the reasoning "the order of the reaction is difficult to assess directly from the shape of the time traces, because the fluorescence signal is a composite of many species contributions". Wouldn't the delay also be a quantity that reflects the behavior of whole sample, because it is determined from that same curve that "is a composite of many species contributions"? I can buy the idea that initially the drain reaction dominates before autocatalysis kicks in, so by assessing the time delay it takes before autocatalysis goes off is an indirect readout of the kinetics of the drain/input dynamics. However, I suggest that these paragraphs are rewritten.

The delay gives an indication of when something happens for the autocatalytic species, whatever it is. On the contrary, the shape of the curve is a composite of fluorescent signals generated by many species (precisely 9), with unknown weights. Fitting the shape of the time trace to derive kinetic information would require to know precisely each contribution to the fluorescent signal, and such an approach would not be robust. On the contrary, C_t values, if much less rich, are robust when it comes to apparent reaction order (as shown by simple math: for example in the case of first order, $[x]=[x_i]e^{kt}$, so the linear relation relating C_t to the log of the

initial concentration value $C_t = 1/k \cdot \ln([x_t]/[x_i])$ holds whatever the threshold value x_t and we do not need to know it to check the reaction order –in practice the threshold has to be small for the kinetic approximations to be relevant).

The paragraph has been modified, see the new version below:

“Mathematically, it can be shown that, just above the critical concentration, the template/drain system should behave as a second-order autocatalytic system (Supplementary Note 3.1). Experimentally, the order of the reaction is difficult to assess directly from the shape of the time traces, because the fluorescence signal is a composite of many species’ contributions. To extract the reaction order, it is more convenient to trigger the system with various initial concentrations $[x_i]$ of its input and observe the time the system takes before it crosses a given small fluorescence threshold (C_t). C_t will relate logarithmically with $[x_i]$ for a first order amplification whereas an inverse law will reveal a second order process. Experimentally, in the absence of drain, we observed regular intervals between the amplification curves for a logarithmic range of initial trigger concentrations, therefore a first-order autocatalytic process. For a system just above the critical drain concentration, this pattern is disrupted and the C_t rather follows an inverse law, symptomatic of a second-order process (Fig. 6). Just below the critical drain concentration we see an intermediate case, which is indicative of an order between 1 and 2. Overall, this confirms that the drain approach, while based on the decay pathway, can be used to change apparent kinetic laws (not rates) of a self-activating positive feedback loop.”

As for the conclusion of this paragraph, it seems to me rather obvious that by introducing new reactions one does not only change the stationary behavior of a system, but also its kinetics.

Again, we wish to emphasize the difference between kinetic rates and kinetic laws. Changing the first is trivial, crafting the second is the art of molecular programming with dynamic circuits. We also wish to emphasize that a much more detailed discussion on this point is provided in the SI Notes 3.1 to 3.3, now better connected to the text as suggested.

- The paper needs to be read more carefully, to make sure that the necessary information is provided timely and accurately to the reader. I found several omitted definitions, and unclear sentences;

We have carefully proofread the paper and made many minor improvements (in red).

for example, at the top of p 8 there is a sentence "results obtained with the other nicking enzyme used in PEN systems (Nt.BstNBI)", that seems completely disconnected from the rest; so far no specific enzyme was mentioned in the main text - prompting the reader to confusion.

This indication has been moved to the discussion, where it now supports the generality of the drain approach, and connected to the corresponding SI:

“We have shown that tinkering with the **kinetic laws of** degradation pathways can be a powerful and general approach to the molecular programming of functional circuits. **While all experiments presented above used Nb.BsmI as the nickase of the PEN machinery, we obtained similar results with the other nicking enzyme used in reported PEN systems (Nt.BstNBI, see Supplementary Note 1.3 and Fig. S2-4).** “

Fig 2 and the main text should explain what enzymes are being used; what are the system inputs and what are its outputs or readouts.

We have now modified the Fig. 3 caption to include this information:

“**Experimental implementation.** **Bst (polymerase), Nb.BsmI (nickase) and ttRecJ (exonuclease) are used to drive an autocatalytic template with or without drains. Fluorescence recordings from an intercalating dye are used to follow in real time the amplification process. (A) the autocatalytic template ctoα2 was incubated without trigger...**”

The paragraph in the middle of page 2 (introduction) describing the PEN toolbox is too vague and requires readers to look up earlier papers.

Due to constraints in the length of the manuscript, and as the details of the PEN DNA toolbox, without drains, have been described in at least 5 publications, the present description of the PEN toolbox was indeed made a bit short. Following this remark, we have added “**Polymerizing/nicking cycles allow the input strand, acting as a trigger, to activate the generation of the signal strand encoded by the output side of the template.**” in order to make the introduction easier to understand.

The time C_t it takes for the system to spontaneously switch steady state from low to high is used early in the paper (Fig 2), but it is only loosely defined at the end of page 8. Please provide a clear explanation of how this time was computed.

We have now defined C_t in Fig. 3 caption and at the beginning of page 7 (the first time C_t are used)

“For each sample, the amplification delay (C_t , set as the time the fluorescence reaches 20% of its normalized maximum) is extracted”

and

We define C_t as the time the untriggered system takes to cross a given small fluorescence threshold (20% of the maximum signal amplitude).

- The supplement should be cleaned up and better connected to the main paper.

We have done some reorganizing and it is now much better referenced in the main text. All SI Notes are called at least once in the main text.

References in the supplement are not compiled and all appear as question marks.

Done

Reviewer #2 (Remarks to the Author):

Question 1:

Authors describe "see SI Appendix for a mathematical discussion" in page 5, but do not show section number or figure number.

Authors should point where the discussion about "additional degradation path" and "small bottleneck" is in "SI appendix for mathematical discussion".

This particular point has been modified in the main text to send the reader to the right section of the supplementary information:

“In other words, the additional degradation path should be fast but have a small bottleneck (see Supplementary Note 3.1).”

Additionally, the corresponding part of the SI has been clarified.

Question 2:

Authors draw regression lines in figure 3-B. However, the number of measurement point is too few,

for example, the regression line of drain-alpha 4 seems to be calculated by using single measurement value.

Authors should measure the $1/C_t$ at more than three drain concentrations for each line in order to verify that the values of $1/C_t$ are lined on the straight lines. In addition, authors should add error bars in all plots.

This point raised by the reviewer is relevant for some, but not all, plots as detailed below.

For Fig. 3A, we want to prove the existence of a bifurcation demarcating the monostable and bistable region, i.e. that the delay time C_t diverges hyperbolically. This translates in a linear relationship between the drain concentration and the inverse of the delay time ($1/C_t$). It is therefore the shape of the curve (the fact $1/C_t$ decreases linearly up to the critical drain concentration, then is at 0) that is important and not so much the quantitative values. This is shown unambiguously by the plot. It is built from a single data point per condition, so it would be inappropriate to draw error bars here³. However a number of other plots showing the same linear relation, thus confirming the trend, are shown in the SI.

The purpose of Fig. 3B is very different. here, we use the linear relationship **demonstrated** above to extract estimates of the critical drain concentrations for various experimental designs. As the drain α 4 is very efficient, the bistability is reached at low concentrations of drain, which explains why there are only 2 data points -not one(!)- that give non-zero $1/C_t$ value for this drain. Although it is indeed not ideal to perform a linear regression with just 2 points, here we are just interested in obtaining an estimation of the critical drain concentration for each drain, and the best estimation possible given the data is the linear extrapolation presented in the plot. In this case, we can easily give a best estimation of the uncertainty of the value, which is simply the interval between the last concentration of drain for which the amplification is observed and the first bistable point. These error bars are now plotted in the inset, where the linear extrapolations are used to give an estimation of the critical drain concentration within this interval.

This procedure is now clearly explained in the caption:

“Figure 3 | Experimental implementation. [...]The critical drain concentration is estimated by extrapolation and reported in the inset with an error bar corresponding to the highest concentration of drain for which spontaneous amplification is observed and the first bistable point, respectively. (C) Influence of Δ_{bind} on the drain’s inhibitory capacity. Autocatalytic templates α_1 , α_2 , α_3 or α_4 (schematized in the upper panels) were incubated in the presence of varying concentrations of a drain template (drain α_1 , concentrations indicated in the colored boxes). Δ_{bind} ranges from 0.2 to 2.4 kcal/mol.”

The other figure mentioned by the reviewer is probably Fig. 6. Again, here we have a single experiment per data point and we are not so much interested in the quantitative value for each point, but rather by the qualitative relation linking the variables depending on various hypotheses concerning the order of the reaction. In this case, the R^2 value gives an appropriate evaluation of the quality of the linear regression and thus is a proper statistical measure of certainty when deciding whether the autocatalytic amplification has an apparent 1st or 2nd order.

Question 3:

In figure 5, authors use dye-labeled template to measure the ON and OFF state. However, ON and OFF state should be defined by the amount of produced DNAs rather than template duplex. So, authors should show the relationship between the fluorescent signals and the amount of the product by direct quantification of the product by other method like qPCR.

It has been previously shown that the concentration of a cognate input is monotonously linked to the fluorescence shift of the dye-labeled template (referred to as N-quenching, Padirac et al. NAR, 2012). Other effects, such as temperature equilibration at the beginning of the reaction or small shifts due to sample injection, do affect, although marginally, the fluorescence signal of the dye. However, the Cy3.5 signal still provides a reliable readout of the state of the switch. We can see for example in Fig. 5 that the signal comes back very close to its initial OFF state after each excursion. Besides the Evagreen signal (which provide an unspecific signal, i.e an evaluation of the total amount of double stranded DNA in the solution at a particular time) is fully consistent. There is therefore no doubt as to when the concentration of the signal species is high or low. In this case, we do not obtain quantitative information about the concentration of gamma during the reaction (note that such quantitative information is provided for the

experiment of Fig. 7), but we can still conclude about the ON/OFF state of the switch by observing the fluorescence shifts.

We have changed the text to better explain this part:

“In Fig. 5, we use fluorescence signals from both the unspecific intercalating dye, and a specific N-quenching modification attached to the template to monitor unambiguously the state of the switch in real time. Repetitive injection of trigger of γ -or degradable inhibitor drainy'- shows that this system can robustly maintain a single bit of memory as the high or low concentration level of a dynamic species, while allowing multiple ON and OFF switching over 15 hours (see Methods for details on fluorescence reporting and analysis).”

Question 4:

Author describe that two independent bistable switches can make four stable states in figure 7-B. I expect that "no trigger" and "beta" shows equivalent Cy3.5 signal because the "beta" does not interact with the gamma-template which is labeled with Cy3.5. Similarly, the value of the beta bistable switch in figure 7-E increases in the presence of other triggers. I'm confused because authors describe "independent" but the results seems not to be independent. Authors should mention about this conflict.

This interesting comment is connected to the question 3, but requires some additional explanations:

First, the N-quenching strategy used to monitor the 2 bit-encoded network (Fig. 7 A-B) allows us to monitor the state of each node. Ideally, the dye-labeled template should respond solely to its cognate trigger. However, it is possible that a slight fluorescence shift is induced by the presence of other species. Especially, all the three triggers used in this figure (α , β and γ) possess the same 5' sequence (6 nucleotides). This might be enough to induce fluorescence crosstalk at the relatively high concentration expected here (up to hundreds of nanomolars, according to the Fig. 7E). This may explain why the Cy3.5 signal is slightly lower once the β node is “ON” than for the negative control (with no trigger), and reciprocally, for the BMN3 signal slightly shifted once the γ node is active.

The direct quantification performed in Figure 7E provides additional insight. In this case, the increase in concentration of the beta strand when the other switches are ON might be explained by competition mechanisms between the 3 nodes, as underlined by the reviewer 1. For instance, when only beta is ON, the exonuclease is less loaded than when other switches are active, and should degrade beta more efficiently, leading to a lower concentration. However, this phenomenon can be counterbalanced by competition for other resources such as the polymerase or the nickase, making precise predictions difficult. In any case, we do not expect “perfect” positions for the fluorescence shifts or the measured concentrations.

In conclusion to this point, we are aware that competition and fluorescence crosstalk exist in such complex molecular mixture, sharing the same enzymatic processor, and we modified the term “independent” in the Figure 7 caption, which was primarily chosen to characterize the network itself but is not appropriate regarding the underlying chemistry.

Changes:

“A 2-bit system can be constructed by combining two bistable networks with orthogonal sequences, as schematized, and monitored with two fluorescent dyes attached to the templates⁴.

and:

“We thus attempted the construction of a system with four stable states, built from a mixture of 2 orthogonal 1-bit molecular circuits of memory (Fig. 7A).”

and:

“When the four oligonucleotides were combined in a tube, the fluorescence signals indeed suggested four stable states”

Question 5:

Although the delta Gs of the drain templates in table 1 seems to be almost the same, why the trigger concentrations after 200 min are differ in alpha, beta and gamma in figure 7-E? According to authors' claim, I expect that the same binding affinity results in the same kinetics of products, or the same amount of products. Author should explain why the amount of the products differs and experimentally verified the explanation.

This comment from the reviewer gave us the opportunity to clarify the link between the thermodynamic parameters of a network and its kinetics. First, as mentioned in the first

paragraph of the Methods sections, thermodynamic parameters (Δ_{ext} , Δ_{bind} , ...) were calculated using the DINAMelt web server. This server (among others) is widely used in DNA nanotechnology (as mentioned by the reviewer #1) to calculate theoretical enthalpic or entropic contributions for DNA hybridization reactions, but they are just estimates and may differ from the experimental values. More importantly, the global kinetics are controlled not only by binding constants but also by other parameters such as the affinity of the 3 enzymes of the toolbox for the different substrates present in the n-bit systems (Figure 7A,C). This has been elegantly demonstrated by Qian *et al.* (Qian *et al.* NAR 2012, now included in the main text), who compared the efficiency of the exponential amplification reaction (using a polymerase and a nicking enzyme) for more than 300 autocatalytic templates. They conclude that “template sequences with similar thermodynamic characteristics perform very differently”, probably due to some sequence preference of the enzymes. We think that this reason is sufficient to explain the difference in trigger production recorded for different sequences. Note that the post-sampling quantification (Fig. 7E) still demonstrates unambiguously the ON/OFF state of the 3 nodes: only traces of trigger are detected in the OFF state of each node while up to a few hundred nanomolars of trigger are measured for an ON node.

To make this clearer, we changed the text to:

For each initial condition, we measured steady state concentrations in the range 100-400 nM for the bits that had been switched ON, while un-triggered bits yielded only traces values (<1nM). The differences in the levels of ON switches, from bit to bit and state to state, can be explained by the differences in amplification/degradation efficiencies of the various sequences, as well as competition effects between switches. Overall, these results are consistent with the presence of eight distinct stable chemical states, hence 3 independent memory bits accessible from the ad-hoc initial conditions (Fig. 7E).

Question 6:

Authors should add an illustration of the network to figure 8 like figure 3-A or figure 2-B for reader's help to understand the network in detail. In addition, authors should add and compare a control which does not include non-linear system to show the importance of the bistable (non-linear) switch for this excitable molecular network.

According to the reviewer's suggestion we modified the Figure 8 with an illustration of the

network. We also included a “negative” control containing only the autocatalytic template without the drain. The reviewer’s suggestion on that point is particularly useful since this control highlights the necessity of the drain template to build the excitable network (without it, one obtains an oscillatory network). The caption has been modified accordingly:

“Figure 8 | Excitable molecular network. A positive feedback loop based on template $\epsilon\sigma\epsilon$ is connected to the cognate drain ϵ . The sequences are designed so that the amplification process is also linked to a negative feedback reaction driving the consumption of triggers ϵ ⁵. A series of replicates are initialized with different concentrations of trigger (from **tubes #1 to #6**: 0, 1.25, 5, 10, 20, 40 nM ϵ) and only the tube #6 shows a strong excitable response. At $t = 545$ and $t = 1435$ min (blue arrows), triggers below (5 nM) or above (40 nM) the amplification threshold are injected in some of the tubes. Consistently, strong responses observed only above the threshold confirm the excitable character of the system. The inset displays the 2D phase plot of the corresponding fluorescence traces, demonstrating the return to the 0 stable state after the excursion. **The tube #7 contains only the template $\epsilon\sigma\epsilon$ (no drain). It exhibits an oscillatory behavior since the low state is unstable and cannot trap the system at the end of a cycle. This observation demonstrates the potency of the drain-based bistable system to set excitable networks.**”

The main text has also been modified to stress this important control:

“The drain template is necessary to trap the system in the 0 state, and forbid the emergence of oscillatory cycles”

Finally, we want to clarify the difference between our work and Subsoontorn et al. paper mentioned in ref 15: these authors indeed present a single node bistable switch, and they use a production-decay mechanism to implement this molecular memory. In their case the system is based on RNA oligos production and degradation (using an RNA polymerase and a mix of RNAses) whereas in our case, the system is based on DNA oligos production and degradation.

However, the similitude with our work does not extend much further.

The most important difference is that the nonlinearity necessary to obtain the bistable behaviour comes in their case from the very non-linear production curve which, by design, present an inherent sharp "activation threshold" (see plots in Fig2). Quoting the authors: "whether the switch is bistable or monostable depends [...] on the output amplitude and the activation threshold". This is to say that bistability depends on features of the production pathway, not the degradation pathway. This can be clearly seen by comparing their plot 6A right, which neatly correspond to our plot in Fig1D, i.e. it is the strategy that we do not take (we take the one of our fig1E, where production is featureless).

It is true that Subsoontorn et al. use a combination of RNAses, which results in a twisty global degradation curve, but this is not essential to their design, and is more an attempt to avoid uncontrolled accumulation of RNA. A perfectly linear degradation curve, if they could obtain it, would do the job as well, because it could still intersect three times with the sigmoidal production curve. In fact, they show that bistability can also be obtained with only RNase H, in which case the high state is not bounded.

Another important difference is that the degradation mechanism of Subsoontorn et al. is non-specific (all RNA molecules will be processed by the RNAses) whereas in our case, the drain template are specific to a particular node and its concentration directly control the non-linear behaviour. Therefore drain templates can be used to tune individually and specifically the non-linearities of each node in a network. This is an essential asset of our method for the building of large scale networks.

In conclusion I think that it is still fair to say that our work is new in its suggestion to use species-specific degradation/deactivation as a way to craft the non-linear dynamics of molecular programs, as opposed to the more traditional focus on non-linear production pathways.

1. Rondelez, Y. Competition for catalytic resources alters biological network dynamics. *Phys Rev Lett* **108**, 018102 (2012).
2. van Roekel, H. W. H. *et al.* Automated design of programmable enzyme-driven DNA circuits. *ACS Synth. Biol.* **4**, 735–745 (2015).

3. Cumming, G., Fidler, F. & Vaux, D. L. Error bars in experimental biology. *The Journal of Cell Biology* **177**, 7–11 (2007).
4. Padirac, A., Fujii, T. & Rondelez, Y. Quencher-free multiplexed monitoring of DNA reaction circuits. *Nucleic Acids Research* **40**, e118 (2012).
5. Fujii, T. & Rondelez, Y. Predator-prey molecular ecosystems. *ACS Nano* **7**, 27–34 (2013).

Reviewer #1 (Remarks to the Author)

The authors clarified my technical question in a satisfactory manner.

As for my major comments:

- My phrase "tuning kinetic rates" may have been misleading, but the manuscript was pretty clear in its first version - its goal is to tune the shape of the degradation rate, not just the degradation speed. I did understand that.

- Broader impact of the PEN toolbox: my original comment mentioned incompatibility with other DNA nanotechnology tools. The authors mention that steps toward using toehold-mediated strand displacement in the PEN toolbox are being made. Yet, I stand by my comment: it will be very hard to interface this toolbox in a scalable manner with other DNA systems; polymerases and nickases are DNA binding enzymes and it won't be easy to prevent unwanted interactions in circuits with a large number of components or use them in the presence of structures.

My general criticism is however mitigated by the authors' mention of efforts to demonstrate the utility of the PEN toolbox in other contexts (e.g. PCR or control of particle aggregation), which has the potential for success.

Overall: while I found it interesting and well packaged, my level of enthusiasm for this project is not very high. My opinion is that it is better suited for a more specialized journal.

Reviewer #2 (Remarks to the Author)

The authors answered my questions clearly. This manuscript is ready to publish in Nature Communications, but I recommend the authors to add the explanation about fluorescent crosstalk in their answer for question 4 into the manuscript.

Reviewer #1 (Remarks to the Author):

The authors clarified my technical question in a satisfactory manner.

As for my major comments:

- My phrase "tuning kinetic rates" may have been misleading, but the manuscript was pretty clear in its first version - its goal is to tune the shape of the degradation rate, not just the degradation speed. I did understand that.

- Broader impact of the PEN toolbox: my original comment mentioned incompatibility with other DNA nanotechnology tools. The authors mention that steps toward using toehold-mediated strand displacement in the PEN toolbox are being made. Yet, I stand by my comment: it will be very hard to interface this toolbox in a scalable manner with other DNA systems; polymerases and nickases are DNA binding enzymes and it won't be easy to prevent unwanted interactions in circuits with a large number of components or use them in the presence of structures.

My general criticism is however mitigated by the authors' mention of efforts to demonstrate the utility of the PEN toolbox in other contexts (e.g. PCR or control of particle aggregation), which has the potential for success.

Overall: while I found it interesting and well packaged, my level of enthusiasm for this project is not very high. My opinion is that it is better suited for a more specialized journal.

...

Reviewer #2 (Remarks to the Author):

The authors answered my questions clearly. This manuscript is ready to publish in Nature Communications, but I recommend the authors to add the explanation about fluorescent crosstalk in their answer for question 4 into the manuscript.

In our previous response to the question regarding fluorescent crosstalk, we only made minor changes to the manuscript; we have now added the following explanation on page 8:

“The Cy3.5 signal (from γ to γ Cy3.5) was actually slightly lower once the β node was “ON” than

in the absence of trigger, and reciprocally, the BMN3 signal (from β to β BMN3) slightly shifted once the γ node was active. Ideally, the dye-labeled templates should respond solely to their cognate trigger. However, as both triggers used (β and γ) possess the same 5' sequence for 6 nucleotides, it is possible that some reporting crosstalk occurred in the presence of high concentrations of the other species.”